

# Comparisons of eccentric knee flexor strength and asymmetries across elite, sub-elite and school level cricket players

Wade J. Chalker[1], Anthony J. Shield[2], David A. Opar[3] and Justin W.L. Keogh[1,4,5]

[1] Faculty of Health Sciences and Medicine, Bond University, Gold Coast, Australia
[2] School of Exercise and Nutrition Sciences, Queensland University of Technology, Brisbane, Australia
[3] Faculty of Health Sciences, Australian Catholic University, Melbourne, Australia
[4] Sports Performance Research Centre New Zealand, Auckland University of Technology, Auckland, New Zealand
[5] Cluster for Health Improvement, Faculty of Science, Health, Education and Engineering, University of the Sunshine Coast, Sunshine Coast, Australia

Corresponding author
Justin W.L. Keogh,
jkeogh@bond.edu.au

## ABSTRACT

**Background.** There has been a continual increase in injury rates in cricket, with hamstring strain injuries (HSIs) being the most prominent. Eccentric knee flexor weakness and bilateral asymmetries are major modifiable risk factors for future HSIs. However, there is a lack of data relating to eccentric hamstring strength in cricket at any skill level. The objective of this study was to compare eccentric knee flexor strength and bilateral asymmetries in elite, sub-elite and school level cricket players; and to determine if playing position and limb role influenced these eccentric knee flexor strength indices.
**Methods.** Seventy four male cricket players of three distinct skill levels performed three repetitions of the Nordic hamstring exercise on the experimental device. Strength was assessed as the absolute and relative mean peak force output for both limbs, with bilateral asymmetries. Differences in mean peak force outputs between skill level and playing positions were measured.
**Results.** There were no significant differences between elite, sub-elite and school level athletes for mean peak force and bilateral asymmetries of the knee flexors. There were no significant differences observed between bowler's and batter's mean peak force and bilateral asymmetries. There were no significant differences between front and back limb mean peak force outputs.
**Discussion.** Skill level, playing position and limb role appeared to have no significant effect on eccentric knee flexor strength and bilateral asymmetries. Future research should seek to determine whether eccentric knee flexor strength thresholds are predictive of HSIs in cricket and if specific eccentric knee flexor strengthening can reduce these injuries.

## INTRODUCTION

Cricket is a team sport involving a side of 11 players with specialist batters, bowlers and one wicket-keeper. Originating from test match cricket which was typically played over 5 days between two teams with unlimited overs, limited overs cricket (50 overs per team)

and Twenty20 (20 overs per team) games lasting 7–8 h and 3 h respectively are also now widely played (*Cannonier, Panda & Sarangi, 2013*). As a result of these new formats, elite level cricket players are now playing more cricket each year, with the intensity of these games also appearing to increase (*Frost & Chalmers, 2014*).

Injury rates appear to be increasing for elite cricketers in many countries, with hamstring strain injuries (HSIs) often the most commonly occurring injury (*Frost & Chalmers, 2014*; *Orchard et al., 2002*; *Orchard et al., 2011*; *Orchard, James & Portus, 2006*; *Stretch, 2003*). Less injuries are reported for junior compared to senior players, with most junior injuries associated with ball contact (*Finch et al., 2010*). Elite pace bowlers may be at higher risk of HSIs than batters and wicket-keepers (*Orchard et al., 2011*). The pace bowler's greater HSI risk may reflect their greater total running and sprint distances (*Petersen et al., 2010*) as well as the intense loads placed on the front leg during ball release when bowling, whereby the knee flexors act to rapidly decelerate the total body's forward momentum resulting from their run-up. Player position (batter or bowler) and limb dominance (front or back leg) during batting, bowling and throwing may therefore have an impact on HSI risk in cricket. However, the symmetry, or lack thereof of knee flexor function is yet to be documented in cricket, unlike other sports (*Bourne et al., 2015*; *Opar et al., 2015*). Throughout this paper, knee flexors refer to the muscle action of knee flexion resulting from the involvement of the hamstrings (bicep femoris long head and short, semitendinosus and semimembranosus) muscles assisted by the gastrocnemius. When the phrase hamstrings is used in the manuscript, it is generally used anatomically in relation to any injury to the bicep femoris, semitendinosus or semimembranosus muscle.

Risk factors that have been associated with HSIs in the wider sports medicine literature include; increased age, previous hamstring injuries, lack of hamstring flexibility, muscle fatigue and hamstring muscle weakness or asymmetry (*Brooks et al., 2006*; *Croisier et al., 2002*; *Croisier et al., 2008*; *Gabbe, Bennell & Finch, 2006*; *Jonhagen, Nemeth & Eriksson, 1994*; *Opar, Williams & Shield, 2012*; *Orchard et al., 1997*; *Verrall et al., 2001*). In particular, eccentric knee flexor strength deficits and bilateral eccentric knee flexor strength asymmetries have been linked to an increased risk of HSIs (*Brooks et al., 2006*; *Croisier et al., 2008*; *Opar, Williams & Shield, 2012*; *Orchard et al., 1997*). A novel field testing device has recently been developed to measure eccentric knee flexor strength and strength asymmetries via uniaxial load cells during the Nordic hamstring exercise (NHE) (*Opar et al., 2013*; *Opar et al., 2015*). Elite Australian football players with eccentric knee flexor strength below 256 N and 279 N at the start and end of pre-season training, respectively, have approximately a 3 to 4 fold increased risk of sustaining a HSI (*Opar et al., 2015*), further confirming the importance of eccentric knee flexor strength in the prevention of HSIs. Additionally, the elevated probability of sustaining a HSI for older athletes or those with a previous HSI can be offset with greater eccentric knee flexor strength (*Opar et al., 2015*). Recent studies using the novel field testing device in elite Australian football and rugby union present somewhat contrasting results on the predictive ability of between limb asymmetries and eccentric knee flexor strength for HSIs. Elite Australian football players with a between limb strength asymmetry of ≥10% did not have an increased risk of sustaining a HSI (*Opar et al., 2015*). This contrasts with a study involving rugby union

players where a between limb strength asymmetry $\geq$15% and $\geq$20% had an increased risk of sustaining a HSI by 2.4 and 3.4 fold respectively (*Bourne et al., 2015*). Such studies suggest that between limb strength asymmetries may be an issue in some sports but not others. Additional studies have found that asymmetries greater than 10%, 8% and 15% for track and field athletes, Australian football players and soccer players respectively, increase the risk of occurrence of a HSI (*Croisier et al., 2008*; *Heiser et al., 1984*; *Opar, Williams & Shield, 2012*; *Orchard et al., 1997*).

Given the lack of data relating to eccentric knee flexor strength in cricket at any level, the asymmetrical demands of cricket batting, bowling and throwing as well as the increasing prevalence of HSIs in elite cricket, the primary purpose of this investigation was to measure eccentric knee flexor strength and bilateral asymmetries in elite, sub-elite and school level cricket players. The secondary purpose was to determine if playing position and limb role influenced eccentric knee flexor strength or the magnitude of asymmetries.

## MATERIALS & METHODS

### Participants

Seventy four male cricket players (16 elite level, 32 sub-elite level and 26 school level) with at least two years of experience in the sport provided written informed consent before participating in the study. Athletes with a previous and/or current lower limb injury that had not yet been fully rehabilitated were not included in the study. Ethical approval was granted by the Bond University Human Research Ethics Committee (RO1824).

### Research design

A cross sectional comparative study design was used with all testing performed at the start of pre-season training before the commencement of a training session, at the start of the training week. This was to be consistent with previous studies using the experimental device (*Bourne et al., 2015*; *Opar et al., 2015*) and to control for any potential differences in training that occur between the playing groups over the course of the season. All participants completed a familiarisation session, in which they performed two sets of three repetitions of the NHE on the novel field testing device while being provided with coaching cues for correct technique. One week later, they were reassessed to determine eccentric knee flexor strength and bilateral asymmetries on a novel field testing device with established reliability (*Opar et al., 2013*). Prior to the main testing session, a submaximal warm-up set of the NHE of three repetitions (first repetition at ~50% of maximum perceived exertion, second repetition at ~70% of maximum perceived exertion and the third repetition at ~90% of their maximum perceived exertion) was performed before the athletes completed one set of three maximal efforts with one minute rest between sets and 15 s inter-repetition rest.

### Eccentric knee flexor strength assessment

Strength was assessed as the peak force output on the field testing device previously assessed for reliability (*Opar et al., 2013*). All participants were asked to kneel on a padded board with their ankles secured immediately superior to the lateral malleolus. Separate securing braces were attached to custom made uniaxial load cells (Delphi Force Measurements)

fitted with wireless data acquisition capabilities (Mantracourt Electronics Ltd). Ankle braces were secured to the testing device via a pivot system to ensure force was always measured through the long axis of the load cells. Athletes were instructed to gradually lean forward at the slowest possible speed while maximally resisting the tendency to fall with both limbs, keeping the trunk and hips in a neutral position throughout the movement with hands placed across the chest (*Opar et al., 2013*). Verbal encouragement and technique coaching was provided throughout each repetition to promote maximal efforts.

### Data analysis

Force data for both left and right limbs during the NHE were logged to a personal computer at 50 Hz through a wireless receiver (Mantracourt Electronics Ltd, Exeter, UK). Mean peak force (N) was calculated for both limbs for the three maximal repetitions. The relative peak force was calculated by dividing the mean peak force (N) by the participant's body mass (kg) so to account for any differences on body-mass between participant groups. The between limb imbalance in eccentric knee flexor force was calculated as a left:right limb ratio of the mean peak force as recommended, using log transformed raw data followed by back transformation (*Impellizzeri et al., 2008*).

### Statistical analysis

All statistical analysis was performed using SPSS version 20.0 (IBM corporation). Means and standard deviations (SD) of age, height, body weight, absolute and relative eccentric knee flexor strength and eccentric knee flexor strength imbalances are presented. If data was not normally distributed, as assessed by Shapiro–Wilks test for normality, log transformation was performed. Log transformed data was then back transformed to represent true values. Data was compared between all three playing groups (elite, sub-elite and school) and further compared based on their self-nominated playing specialisation (bowler or batter) with Cohen $d$ values provided. Small effect sizes were classified as >0.2, moderate effect size >0.5 and a large effect size ≥0.8. One way analysis of variance with a Tukey post hoc test was used to compare mean peak force output (combined average of left and right limbs) and limb asymmetry between the three groups (elite, sub-elite and school). An independent $t$-test compared the mean peak force and limb asymmetry between playing positions (bowlers and batters). Paired $t$-tests were performed comparing front and back leg mean peak force outputs, where front leg was considered the leg most forward at the point of ball release when bowling and throwing and in the normal batting stance for batters. Within-session variability of peak force and limb asymmetry were quantified via the coefficient of variation, calculated for all three groups based on the three maximal repetitions. Significance was set at $P < 0.05$.

## RESULTS

The physical characteristics of each skill level and playing position are described in Table 1. Increases in skill level were associated with significantly increased player age, height and weight.

There were no significant differences between elite, sub-elite and school level athletes for absolute mean peak force (elite vs sub-elite $d = 0.07$; sub-elite vs school $d = 0.32$; elite vs

**Table 1  Physical characteristics of participants in each skill level and playing position.**

|  | Elite ($n = 16$) | Sub-elite ($n = 32$) | School ($n = 26$) | Combined ($n = 74$) |
|---|---|---|---|---|
| Age | $24.5 \pm 4.5$ | $21.5 \pm 4.2^{*}$ | $15.7 \pm 1.0^{*,**}$ | $20.1 \pm 4.9$ |
| Height (cm) | $186.2 \pm 9.5$ | $184.2 \pm 6.9$ | $178 \pm 7.2^{*}$ | $183.1 \pm 7.9$ |
| Weight (kg) | $86.3 \pm 8.1$ | $83.1 \pm 10.3$ | $69.4 \pm 11.0^{*,**}$ | $79.0 \pm 12.3$ |

|  | Bowlers ($n = 42$) | Batters ($n = 32$) |
|---|---|---|
| Age | $20.1 \pm 4.7$ | $20.1 \pm 5.2$ |
| Height (cm) | $184.2 \pm 8.4$ | $181.7 \pm 7.1$ |
| Weight (kg) | $80.3 \pm 13.0$ | $77.3 \pm 11.3$ |

**Notes.**
[*]Significantly different to elite level athletes.
[**]Significantly different to sub-elite level athletes.

school $d = 0.42$; $P > 0.05$), relative mean peak force (elite vs sub-elite $d = -0.10$; sub-elite vs school $d = -0.42$; elite vs school $d = -0.55$; $P > 0.05$) and bilateral asymmetries (elite vs sub-elite $d = -0.34$; sub-elite vs school $d = 0.21$; elite vs school $d = -0.11$; $P > 0.05$)(see Table 2). 24 athletes (32.4%) had bilateral limb asymmetries $\geq 15\%$, and 14 (18.9%) of these athletes had asymmetries $\geq 20\%$. Furthermore, Table 1 indicates there were no significant differences observed between bowler's and batter's absolute ($d = -0.11$; $P > 0.05$) and relative mean peak force ($d = -0.29$; $P > 0.05$) and bilateral asymmetries ($d = 0.04$; $P > 0.05$). Of the 24 athletes with limb asymmetries $\geq 15\%$, 14 (33.3%) were bowlers and 10 were batters (31.3%). Nine bowlers (21.4%) and 5 batters (15.6%) had limb asymmetries $\geq 20\%$. There were no significant differences between front and back limb absolute mean peak force outputs ($299 \pm 79$ N and $303 \pm 71$ N; $d = -0.05$; $P > 0.05$) and relative mean peak force ($3.83 \pm 1.03$ N kg$^{-1}$ and $3.84 \pm 0.84$ N kg$^{-1}$; $d = -0.01$; $P > 0,05$) (see Fig. 1).

## DISCUSSION

To the best of our knowledge, this is the first paper to measure eccentric knee flexor strength in cricket players of any level. As HSI rates are increasing in cricket (*Frost & Chalmers, 2014*; *Orchard et al., 2011*; *Orchard, James & Portus, 2006*), this study may present the first step in identifying the potential role of eccentric knee flexor strength and bilateral asymmetries in predisposing cricket players to such injury. The lack of any significant difference in eccentric knee flexor strength or asymmetry between skill level was surprising. Elite players were expected to be stronger as they should have greater experience in performing knee flexor strengthening exercises and they place greater loads on their hamstrings as a result of their greater running and sprinting distances (*Petersen et al., 2010*). There was only a small effect size between sub-elite and school level athletes and a small to moderate effect size between elite and school level athletes for absolute mean peak force. Relative mean peak force demonstrated a small to moderate effect size between sub-elite and school level athletes and a moderate effect size between elite and school level athletes. While cricket players possessed $\sim 80\%$ of the eccentric knee flexor strength of rugby union players of similar levels (367 N, 389 N and 343 N for elite, sub-elite and under 19 athletes respectively)

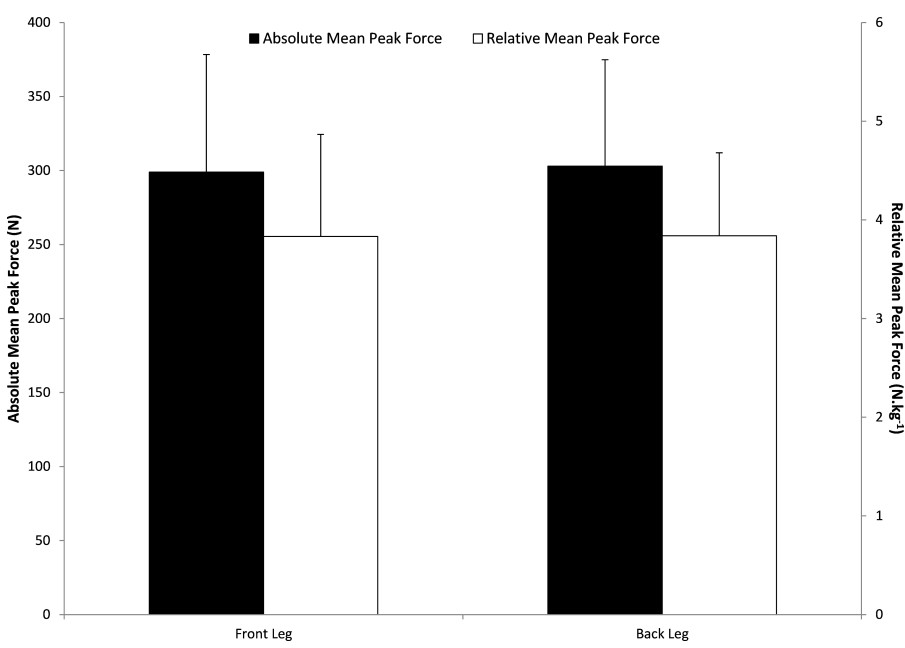

**Figure 1** Mean peak force outputs (N) and relative mean peak force outputs (N kg$^{-1}$) for combined group ($n = 74$) comparing front leg to back leg.

(*Bourne et al., 2015*). The differences in eccentric knee flexor strength between skill levels of rugby union players appears similar in magnitude to that of the current study in cricket, however, there was a significant difference in bilateral eccentric knee flexor force outputs between the sub-elite and under 19 athletes in rugby union. Therefore, differences between the cricket groups for mean peak force were expected considering the mean age of the school athletes (15.7 ± 1 year) is lower than the rugby under 19 athletes. Due to the greater eccentric knee flexor demands in elite compared to sub-elite and schoolboy sport, the results of this study and that of *Bourne et al. (2015)* may suggest that additional specific eccentric knee flexor exercises may be required in elite cricket and rugby to reduce their high prevalence of HSIs. Support for this approach can be found in soccer where the addition of the NHE significantly reduced HSI rates (*Arnason et al., 2008*; *Askling, Karlsson & Thorstensson, 2003*). Further, the fact that the school level athletes possess greater relative mean peak force compared to elite and sub-elite athletes may explain why the occurrences of HSIs in junior cricket are much lower compared to elite athletes who are playing and training under greater physical demands with a reduced capacity to protect the knee flexors.

Compared to elite batters, elite pace bowlers may be more prone to HSIs due to the high-intensity eccentric loads placed on the hamstrings of the front leg at ball release during bowling (*Orchard et al., 2011*) and their greater sprinting distances (*Petersen et al., 2010*). This suggests that elite pace bowlers may be the playing group most in need of additional eccentric knee flexor strengthening exercises. Support for this view is provided by our findings which showed no significant difference in eccentric knee flexor strength or bilateral asymmetries between bowlers and batters. Additionally, bowlers were found to have less relative mean peak force compared to that of batters (3.74 ± 0.97 N kg$^{-1}$
**Table 2** Mean peak force outputs and relative mean peak force outputs with bilateral limb asymmetries of knee flexors during the NHE for all three groups; elite, sub-elite and school level and playing positions; bowlers and batters.

| | Absolute mean peak force (N) | Relative mean peak force (N kg$^{-1}$) | Bilateral asymmetry (%) |
|---|---|---|---|
| Elite | $313 \pm 67$ | $3.65 \pm 0.89$ | $11.5 \pm 8.6$ |
| Sub-elite | $308 \pm 77$ | $3.74 \pm 0.96$ | $15.1 \pm 12.2$ |
| School | $285 \pm 68$ | $4.11 \pm 0.77$ | $12.6 \pm 11.6$ |
| Bowler | $297 \pm 77$ | $3.74 \pm 0.97$ | $13.7 \pm 10.3$ |
| Batter | $305 \pm 65$ | $3.99 \pm 0.76$ | $13.2 \pm 12.5$ |

**Notes.**
Data presented as mean $\pm$ standard deviation.

and $3.99 \pm 0.76$ N. kg$^{-1}$, respectively), where it is believed they should maintain greater relative force as they are exposed to more eccentrically demanding knee flexor actions. The relatively low average bilateral asymmetry for both playing positions suggests that the game is not subjecting athletes to chronic limb asymmetries. However, greater consideration for eccentric knee flexor strengthening exercises may need to be implemented for bowlers at an elite and sub-elite level to help combat the high incidence rates of HSIs.

It is however unclear whether such additional eccentric knee flexor training should focus on reducing bilateral asymmetries or increasing the strength of both knee flexors in elite cricketers. Elite Australian football players with an eccentric knee flexor force output below 256N at the start of pre-season training were at an increased risk of sustaining a HSI (*Opar et al., 2015*), within the 74 participants of this study, 21 (28.4%) had a mean peak force output below 256N, and when separated into playing positions, 14 bowlers (33.3%) and 7 batters (21.9%) had mean peak force outputs below 256N. Elite Australian football players with a bilateral limb asymmetry of up to 20%, as measured on the same testing device as the current study, did not increase the risk of sustaining a HSI (*Opar et al., 2015*). This contrasts with rugby union, where a 2.4 and 3.4 fold increase of HSI risk occurred with asymmetries of $\geq$15% and $\geq$20% respectively (*Bourne et al., 2015*). Results of the current study indicated similar mean levels of bilateral limb asymmetries for bowlers (13.7%) and batters (13.2%) and across all three skill levels (11.5–15.1%). This is further supported with similar proportions of athletes at each skill level with bilateral limb asymmetries $\geq$15% (31.3%, 34.4% and 30.8% for elite, sub-elite and school level groups, respectively). While the contribution of knee flexor strength asymmetry on HSI rates seems to differ between sports, further prospective studies involving cricket players throughout an entire season may be required to better understand this possible relationship.

There was no significant difference between absolute and relative mean peak force for front limb and back limb. It was expected that front limb would be stronger in both playing positions as batters associate their front leg with the greatest amount of use and bowler's front legs are associated with the greatest eccentric demands during the bowling action at ball release. The exact causes for similar front and back limb forces across all participants are unknown and it may come down to multiple influences that differ between each player.

To some extent the current study is limited by the sample sizes for each group, whereby larger samples would allow greater certainty of the effect of skill level, playing position and leg dominance on knee flexor limb strength and asymmetries. The cross-sectional nature of the data also limits the inferences that can be drawn from the data. It is acknowledged that the time of testing may not truly represent playing positions, however, it is believed as all athletes are experienced in the game of cricket and their playing position any chronic asymmetries between limbs would likely still be present at the start of pre-season training.

## CONCLUSION

There appears to be no significant differences between eccentric knee flexor strength or between limb strength asymmetries for elite, sub-elite and junior players during the NHE. Further, playing position (batter or bowler) or leg role (front or back) appears to have no significant effect on eccentric knee flexor strength and between limb strength imbalances for cricket players. Further research is required to prospectively monitor cricket players to determine if eccentric knee flexor strength or between limb strength asymmetries is a predictor of hamstring injury and if eccentric knee flexor strengthening exercise programs may reduce injury rates in cricket.

### Funding
The authors received no funding for this work.

### Competing Interests
Anthony J. Shield and David A. Opar are listed as co-inventors on an international patent application filed for the experimental device (PCT/AU2012/001041.2012).

### Author Contributions
- Wade J. Chalker conceived and designed the experiments, performed the experiments, analyzed the data, wrote the paper, prepared figures and/or tables.
- Anthony J. Shield and David A. Opar conceived and designed the experiments, contributed reagents/materials/analysis tools, reviewed drafts of the paper.
- Justin W.L. Keogh conceived and designed the experiments, reviewed drafts of the paper.

### Human Ethics
The following information was supplied relating to ethical approvals (i.e., approving body and any reference numbers):
Bond University Human Research Ethics Committee, Gold Coast, Australia; RO1824.

### Data Availability
The raw dataset was supplied as Data S1.

## Supplemental Information

Supplemental information for this article can be found online at http://dx.doi.org/10.7717/peerj.1594#supplemental-information.

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
