# Peer review of "Comparisons of eccentric knee flexor strength and asymmetries across elite, sub-elite and school level cricket players"

_PeerJ, doi:10.7717/peerj.1594_

## Round 0.1 · original submission · Major Revisions

· Academic Editor

Major Revisions

I have a few comments to add in addition to those of the reviewers.

1) given that there were no differences in your measurements, I suggest the title of the manuscript be reworded appropriately. In its current form the title suggests that there are differences in KF strength and asymmetries.

2) Please be consistent with the use of knee flexors and hamstrings, they are used interchangeably throughout the manuscript.

3) I found it very surprising, as you mention, that there were no statistically significant differences in the KF strength between the different groups. The elite players, for example, are significantly bigger and older that the school aged participants. Given the type of measure that was used, would it be more appropriate to use relative strength measures and if so what would this mean (the school aged participants would be 'stronger')? A similar comment is made by Reviewer #2.

4) As pointed out by Reviewer #1, it may also be very useful to report effect sizes given the applied nature of this study and its potential implications.

Please carefully consider the comments from both reviewers. I shared many of their concerns and points of clarifications.

·

Basic reporting

Introduction

I personally found that a short paragraph explaining the influence of bilateral asymmetries is lacking in the introduction. For example, from what % differences does the risk of injury increase? This is especially the case when considering the relatively long explanation on the importance of peak eccentric strength and its relation to injuries (line 48-53).

Experimental design

Research design

Line 75- What is the definition of a sub-maximal NHE? How is this performed? How is the duration of each repetition, if at all, controlled for?

Line 120 – While no statistical differences were found between level of athletes, both elite and sub-elite athletes are considerably stronger than the school level athletes. Indeed, a comparison between elite and school level athletes using Cohen’s d effect size led to a small to medium differences equal to 0.41, and a small effect size equal to 0.31 between sub-elite and school level athletes. I suggest mentioning these differences as they are meaningful.

Validity of the findings

Discussion

In line 138 it is written that the lack of differences between the level of athletes in the present study is surprising, yet in line 143 a study that made a similar comparison between Rugby players is cited which found similar results to the present study. That is, no differences. Considering the lack of differences between the athletes in the Rugby study, why are the results of the present study surprising? Also, it may be appropriate to discuss the previously mentioned effect size differences in this section.

Line 147- While elite athletes may require greater eccentric forces compared to sub-elite and school level athletes, I wonder if they are more prone to such injuries. A discussion of the required forces between levels of athletes in view of injury frequencies is warranted in my opinion. Furthermore, it would be interesting to discuss the findings in view that maximal eccentric strength may not be a large risk factor for future HSI in Cricket (provided that it is above the required baseline). Indeed, the lack of differences between the groups may point to other factors, such as accumulated fatigue, and weekly training hours, that may be of greater relevance to future injuries provided that, for example, peak eccentric forces are above the minimum criteria of 255 N and the bilateral asymmetries are smaller than 20%.

Comments for the author

I found the article to be well written and conducted in sound manner. The research question is a relevant one, and while the results are surprising, they further our understanding of potential causes of hamstring injuries.

·

Basic reporting

I agree with the authors in that this study, by identifying differences (or in the case lack of differences) in eccentric knee flexor strength and bilateral asymmetries between skill level, position, and limb role, is a logical “first step”. The authors need to keep in mind the stated objectives of their study and be careful of extending their findings to far (i.e. “risk of injury” in cricket players). For example, in the discussion section (L 152-158), why did playing position have no influence on peak eccentric strength and bilateral asymmetry? The authors discuss the implications (risk of injury, training needs) of the lack of difference, but do not offer up much in terms of an explanation. The same applies for the influence of limb role. These sections of the discussion need to be addressed.

The authors have identified a knowledge gap and have provided sufficient background in identifying a clear and important research question. The rationale for this study is well established.

Please carefully proof the manuscript for editing errors. For example (Line 2 of abstract), “Eccentric knee flexor weakness and bilateral asymmetries are a major modifiable risk factor for future HSIs.” Should be “Eccentric knee flexor weakness and bilateral asymmetries are major modifiable risk factors for future HSIs.”

Experimental design

Given the stated objectives, the experimental design and methods are appropriate.

With that said, there are a couple of concerns the authors should address:
1. The time of season of testing (i.e. preseason) – The secondary objectives of this study were to examine playing position and limb role on peak eccentric hamstring strength. If the testing took part in the preseason, meaning players have not played in their identified playing position, the results wouldn’t seem to be reflective of the influence of playing season and limb role. Please address.
2. Do the researchers have any concerns with using a bilateral test to measure unilateral strength to then determine a left vs. right strength asymmetry? In other words, is a bilateral test (in this case the Nordic hamstring exercise [NHE]) a valid assessment of unilateral strength (as an example, any bilateral strength deficit in the players would mean the measures of unilateral strength via the NHE are less than “actual” unilateral strength? Why not perform a unilateral test for the right and left hamstring muscle group?

Validity of the findings

What is the basis for the “cutoff values” (lines 121-123, e.g. 256 N)? These seem to apply to sports other than cricket (and contact sports nonetheless). Comparison to these cutoffs may be a better fit in the discussion. Otherwise, define/explain your cutoff values in the methods section.

Have the authors considered expressing force relative to body mass? Given that bowlers need to decelerate their own body mass, this may be a useful value. Rough calculations would suggest the relative peak eccentric strength for elite, sub-elite, and school players would be 3.6, 3.7, and 4.1 N/kg. Although purely speculation at this point, perhaps the drop in relative strength (high school vs. elite level) may be associated with the greater hamstring injury incidence in seniors vs junior aged athletes as suggested by Finch et al. 2010.

Comments for the author

No other comments

---

## Round 0.2 · accepted · Accept

· Academic Editor

Accept

Dear Dr. Keogh,

It is a pleasure to accept your manuscript entitled, "Comparisons of eccentric knee flexor strength and asymmetries across elite, sub-elite and school level cricket players" in its current form for publication in PeerJ.

·

Basic reporting

No Comments

Experimental design

No Comments

Validity of the findings

No Comments

Comments for the author

I am happy with the changes. Congratulations.

·

Basic reporting

No new comments

Experimental design

No new comments

Validity of the findings

No new comments

Comments for the author

The authors have addressed all my initial concerns.